# Development of a Nanoparticle-Based Approach for the Blood–Brain Barrier Passage in a Murine Model of Amyotrophic Lateral Sclerosis

**DOI:** 10.3390/cells11244003

**Published:** 2022-12-10

**Authors:** Martina Bruna Violatto, Laura Pasetto, Elisabetta Casarin, Camilla Tondello, Elisa Schiavon, Laura Talamini, Gloria Marchini, Alfredo Cagnotto, Annalisa Morelli, Alessia Lanno, Alice Passoni, Paolo Bigini, Margherita Morpurgo, Valentina Bonetto

**Affiliations:** 1Department of Biochemistry and Molecular Pharmacology, Istituto di Ricerche Farmacologiche “Mario Negri” IRCCS, 20133 Milan, Italy; 2Ananas Nanotech S.r.l., 35131 Padua, Italy; 3Department of Pharmaceutical and Pharmacological Sciences, University of Padova, 35122 Padua, Italy; 4Pharmazentrum Frankfurt/ZAFES, Goethe University Hospital Frankfurt, 60596 Frankfurt am Main, Germany; 5Department of Environmental Health Sciences, Istituto di Ricerche Farmacologiche “Mario Negri” IRCCS, 20133 Milan, Italy

**Keywords:** amyotrophic lateral sclerosis, Nanomedicine, pharmacology, blood–brain barrier

## Abstract

The development of nanoparticles (NPs) to enable the passage of drugs across blood–brain barrier (BBB) represents one of the main challenges in neuropharmacology. In recent years, NPs that are able to transport drugs and interact with brain endothelial cells have been tested. Here, we investigated whether the functionalization of avidin-nucleic-acid-nanoassembly (ANANAS) with apolipoprotein E (ApoE) would allow BBB passage in the SOD1^G93A^ mouse model of amyotrophic lateral sclerosis. Our results demonstrated that ANANAS was able to transiently cross BBB to reach the central nervous system (CNS), and ApoE did not enhance this property. Next, we investigated if ANANAS could improve CNS drug delivery. To this aim, the steroid dexamethasone was covalently linked to ANANAS through an acid-reversible hydrazone bond. Our data showed that the steroid levels in CNS tissues of SOD1^G93A^ mice treated with nanoformulation were below the detection limit. This result demonstrates that the passage of BBB is not sufficient to guarantee the release of the cargo in CNS and that a different strategy for drug tethering should be devised. The present study furthermore highlights that NPs can be useful in improving the passage through biological barriers but may limit the interaction of the therapeutic compound with the specific target.

## 1. Introduction

In the field of neurology, a relevant number of encouraging results have been described from in vitro studies, generating a number of new potential drug candidates. Nevertheless, translating the results from in vitro to in vivo or from preclinical studies to clinical trials remains very challenging. One of the main hurdles is the limited passage of drugs from the circulatory system to the central nervous system (CNS). To prevent the risks deriving from an uncontrolled passage of potentially hazardous substances, the blood–brain barrier (BBB), formed by brain endothelial cells, strictly segregates blood components, drastically reducing the entry of circulating compounds inside the CNS. By this mechanism, the BBB also limits the cerebral uptake of therapeutic molecules. The main factors that influence BBB permeability include molecular weight, charge, lipid solubility, surface activity, and the relative size of the compounds [1]. Over the years, great efforts have been taken to deliver drugs and diagnostic agents to the brain. A series of technical approaches have been tested, among which are the use of viral vectors, nanoparticles (NPs), extracellular vesicles, and/or by trying to exploit active transporters, brain permeability enhancers, or non-invasive techniques (ultrasounds, magnetic stimulations). Unfortunately, the results are often controversial and lack the robustness to further proceed to clinical application. In some pathological conditions, the BBB has a higher permeability, making the passage of therapeutic compounds easier. In particular, neuroimaging studies have demonstrated BBB dysfunction in many neurodegenerative diseases, including amyotrophic lateral sclerosis (ALS) [2,3].

Nanotechnology could play a pivotal role in the development of strategies for drug delivery in ALS [4,5]. In this context, different types of materials (such as lipid-based, polymeric, and inorganic NPs) have been engineered to deliver therapeutics to the brain. NPs can be engineered to display specific physicochemical features to favor drug delivery, for example, by increasing bloodstream stability, reducing drug clearance, or improving affinity to the cellular target [6].

In order to increase the affinity of NPs to brain capillaries, surface modification is greatly exploited. Brain capillary endothelial cells express a large number of blood-to-brain transport systems that, in principle, could facilitate the entry of compounds into the brain. Peptides and proteins, such as insulin, transferrin, or lipoproteins, are indeed transported across the BBB via receptor-mediated transcytosis. Apolipoprotein E (ApoE) associates with lipids to form lipoproteins and plays a key role in the transport and uptake of cholesterol to the brain [7]. Lipoprotein-associated receptors are highly expressed by brain endothelial cells, and ApoE has been extensively investigated as a major ligand candidate for NP-based drug delivery targeting the brain [8]. In fact, it was shown that ApoE guided the NP’s display and increased endocytosis and transcytosis, especially in vitro.

An attractive alternative NP platform combining advanced surface chemistry to a safe interaction with biological targets is represented by the nanoassembly ANANAS. ANANAS are poly-avidin NPs that form upon the high affinity-driven nucleation of avidin units around a non-coding plasmid DNA [9,10]. Each nanoparticle, which is protected by a PEG layer for colloidal stability (Figure 1), possesses a large number of biotin binding sites (BBS), which are available for docking about 1000 functional elements (drugs, fluorophores for tracking and/or targeting elements), provided these are linked to a biotin moiety. The high affinity of biotin for avidin (Kd~10^−15^ M) permits the exploitation of the available BBS to obtain functional NPs with stoichiometric control of composition by simply mixing core NPs (which are obtained and freeze-dried in a separate process) with the desired biotinylated moieties. Functional NPs are thus obtained in a “one pot” solution, and as long as the number of available BBS or the available NP surface (about 6000 nm^2^) is not exceeded, they can be used without the need for purification.

In the past few years, these protein-based assemblies have been proposed as platforms for drug delivery, showing great potential for the treatment of cancer [11] and liver disease [12]. ANANAS are extremely promising because of their safety, multifunctionality, stoichiometric drug encapsulation, defined composition, and scalability but also for their high biocompatibility, biodegradability, and low immunogenicity [13].

In the present study, we exploited the ANANAS platform to investigate NP delivery to the CNS in wild-type (WT) and SOD1^G93A^ mice, which develop a motor neuron disease and display morphological damages to BBB and the blood-spinal cord barrier (BSCB) as the disease progresses [14]. To evaluate the ability of intravenously injected ANANAS and ANANAS-ApoE to reach CNS tissues, we developed an experimental platform combining biochemical and imaging approaches. Finally, to evaluate if the BBB passage could be accompanied by a local release of a therapeutic payload, we treated the SOD1^G93A^ mice with an ANANAS-based nanodrug [12,15] carrying the steroid dexamethasone (Dex), and we evaluated the amount of Dex released in the brain and spinal cord. Dex was selected as a potential drug candidate to control the neuroinflammatory response associated with motor neuron diseases [16,17]. Targeting steroids in the CNS by means of nanocarriers may be a way to maximize their therapeutic effect. In the ANANAS-Dex here used, the drug was tethered through an acid-reversible hydrazone bond [12,15], which should be capable to release the free drug upon reaching an acidic compartment, such as an inflamed tissue or a target cell endosomal/lysosomal compartment.

## 2. Materials and Methods

### 2.1. Materials

Avidin was purchased from e.protein (Belgium), biotin-methoxy-PEG5kDa (B-mPEG5kDa), and biotin-PEG5kDa-NH_2_ were purchased from Laysan bio (Arab, AL, USA). Atto488-NHS was purchased from Attotech Gmbh (Siegen, Germany, code # 11U26). Biotin-C6-Alexa633, Biotin-L-lys-(methyl-PEG5kDa)_2_, and biotin-C6-Cb-Hz-Dexamethasone (B-C6-Cb-Hz-Dex) were synthesized according to published procedures [10,13,15]. Maleimidosuccinimidyl propionate (MSP) and all other reagents were purchased from Sigma Aldrich. Biotin-PEG5kDa-propylamido-maleimide was obtained by mixing biotin-PEG5kDa-NH_2_(LaysanBio lot #127-123) with 3 equivalents of MSP in 10 mM phosphate, 150 mM NaCl, pH 7.4 (PBS buffer). The product was purified by gel filtration on a G25 resin (NAP10, Cytiva Life Sciences, Karnataka, India) using water as an eluent.

### 2.2. Biotin-PEG5kDa-Cys-ApoE Synthesis

The peptide, corresponding to residues 141–150 of human ApoE CWG-(LRKLRKRLLR), was synthesized on an automated Alstra synthesizer (Biotage, Uppsala, Sweden) at a 0.1 mM scale with NOVASYN-TGA resin (Novabiochem, San Diego, CA, USA) using Fmoc-protected L-amino acids (Sigma Aldrich, St. Louis, MO, USA). The peptide was bearing in the N-term position a glycine residue as a spacer, a tryptofan residue for fluorescence monitoring, and ended with cysteine for covalent coupling with ANANAS. Amino acids were activated by a reaction with O-(Benzotriazol-1-yl)-N,N,N’,N’-tetramethyluronium tetrafluoroborate and N,N-diisopropylethylamine. A capping step with acetic anhydride after the last coupling cycle of each amino acid was included. The peptide was cleaved from the resin with trifluoroacetic acid/thioanisole/water/phenol/ethanedithiol (82.5:5:5:5:2.5 *v*/*v*), precipitated, and washed with diethyl ether. The precipitate was then purified by reverse-phase high-performance liquid chromatography on a semi-preparative C4 column (Waters Corporation, Milford, MA, USA). The correct peak fraction corresponding to the peptide molecular weight was identified using a MALD-TOF spectrometer (Applied Biosystems, Waltham, MA, USA) before being freeze-dried and stored at −20 °C until use. The peptide purity was higher than 95%. Biotin-PEG5kDa-Cys-ApoE was obtained by mixing Biotin-PEG5kDa-propylamido-maleimide with 1 equivalent of ApoE CWG-(LRKLRKRLLR). The product was purified by gel filtration. Peptide coupling to the biotin-PEG-linker was confirmed by a combination of analytical techniques: UV-Vis spectrophotometry was used to measure the ApoE concentration, HABA assay was used to measure biotin content [18], and iodine assays [19] measured the PEG content.

### 2.3. ANANAS Formulations

Core ANANAS carrying the surface protective biotin-L-lys-(methyl-PEG5kDa)_2_ to guarantee colloidal stability were prepared according to Pignatto [10] and were freeze-dried. For functional assembly preparation, core NPs were reconstituted in the buffer, and biotinylated elements were added (Biotin-PEG5kDa-ApoE, biotin-mPEG, biotin-C6-Cb-Hz-Dex, SI for chemical structures) to the desired biotin/biotin binding site (BBS) molar ratios. When needed, biotin-C6-Alexa633 was also added at 15% BBS coverage [13]. NPs were characterized by size by the Zetaseizer nano ZS (Malvern instruments) instrument. The NP loading capability for different biotin elements was measured by a combination of tools, including gel permeation chromatography, dot blot analysis, and DLS (see also Appendix A).

### 2.4. Animals

The “Mario Negri” Institute for Pharmacological Research IRCCS adheres to the principles set out in the following laws, regulations, and policies governing the care and use of laboratory animals: Italian Governing Law (D.lgs 26/2014; Authorization no. 19/2008-A issued March 6, 2008, by Ministry of Health); Mario Negri Institutional Regulations and Policies providing internal authorization for persons conducting animal experiments (Quality Management System Certificate, UNI EN ISO 9001:2015, Reg. No. 6121), the NIH Guide for the Care and Use of Laboratory Animals (2011 edition), and EU directives and guidelines (EEC Council Directive 2010/63/UE). This work was reviewed by the internal Animal Care and Use Committee (IACUC) and approved by the Italian “Istituto Superiore di Sanità” (code: 722/2017-PR). Animals were maintained under specific pathogen-free conditions and regularly checked by a veterinarian responsible for animal welfare supervision and experimental protocol review. Mice were bred in standard conditions: temperature 21 ± 1 °C, relative humidity 55 ± 10%, 12 h light schedule, and food and water ad libitum. Both in vivo and ex vivo analyses were performed on CD1 and SOD1^G93A^ mice. The CD1 mouse (WT mice), acquired from Charles Rivers, is a multipurpose model. It is an albino outbred strain of mouse model that has frequently been used in toxicology (safety and efficacy study) and pharmacological research. The SOD1^G93A^ mouse is an established mouse model of ALS. The SOD1^G93A^ line on a homogeneous 129S2/SvHsd background derives from the B6SJL-TgNSOD-1-SOD1G93A-1Gur line that was originally obtained from The Jackson Laboratory (Bar Harbor, ME, USA), which expresses about 20 copies of mutant human SOD1^G93A^ [20,21]. This transgenic SOD1^G93A^ mouse strain develops the first signs of motor neuron pathology at about 4 weeks of age. Mice start to show muscle strength and motor function impairment at 14 weeks of age and survive for up to 18 weeks of age. Genotyping for SOD1^G93A^ was performed by a standard PCR using primer sets designed by The Jackson Laboratory. The number of animals was calculated on the basis of experiments designed to reach a power of 0.8, with a minimum difference of 20% (α = 0.05).

### 2.5. In Vivo Treatments

The experimental groups enrolled for in vivo and ex vivo analyses are summarized in Table 1. SOD1^G93A^ mice were treated at the onset of symptoms (14 weeks of age), and the same age was maintained for the treatment of WT mice. In detail, WT and SOD1^G93A^ mice were intravenously treated with ANANAS-mPEG (ANANAS) 0.54 mg NPs/mouse (21.6 mg/kg) or ANANAS 0.54 mg NPs/mouse (21.6 mg/kg) loaded with 10% ApoE (ANANAS-ApoE) or PBS as a vehicle. In addition, another group of SOD1^G93A^ mice received intravenously ANANAS 1 mg NPs/mouse (40 mg/kg) loaded with dexamethasone 17.1 µg, Dex/mouse (0.68 mg/kg) (ANANAS-Dex), or free Dex 17.1 µg Dex/mouse (0.68 mg/kg). Mice were sacrificed at different time points (Table 1) and were analysed for pharmacokinetics measurements through biochemical (dot blot, HPLC/MS) and imaging approaches (ex vivo imaging).

### 2.6. Tissue Dissection and Plasma Isolation

Mice were deeply anesthetized with an overdose of ketamine hydrochloride (IMALGENE, 150 mg/kg; Alcyon Italia) and medetomidine hydrochloride (DOMITOR, 2 mg/kg; Alcyon Italia) by intraperitoneal injection. Before sacrifice, all groups except one in a preliminary experiment were perfused transcardially with 50 mL of phosphate-buffered saline (PBS). The liver, brain, and spinal cords were rapidly removed, collected, and frozen at −80 °C for subsequent analysis. Blood samples were collected in EDTA pre-coated vials and centrifuged at 13,400× *g* for 2 min. Mouse plasma samples were stored at −80 °C until used.

### 2.7. Dot Blot Analysis

Tissues were homogenized by sonication in 1% boiling SDS. Protein homogenates were further boiled for 10 min and centrifuged at 13,500× *g* for 5 min. In order to quantify the proteins, supernatants were analysed by the BCA protein assay (Pierce) and subsequently boiled for 1 h immediately before dot blot analyses. The treatment was carried out on tissues before dot blot analysis so as to denature and thus disassemble the ANANAS. This permitted the detection/quantification of their presence in the tissues by titering avidin: their major component. For dot blot, proteins (3 μg) were loaded directly onto nitrocellulose Trans-blot transfer membranes (0.45 μm; Bio-Rad, Hercules, CA, USA) by vacuum filtration, as described previously [22]. Dot blot membranes were blocked with 3% (*w*/*v*) BSA (Sigma-Aldrich, St. Louis, MO, USA) and 0.1% (*v*/*v*) Tween 20 in Tris-buffered saline, pH 7.5, were incubated with primary antibodies and then with peroxidase-conjugated secondary antibodies (GE Healthcare, Chicago, IL, USA). Antibodies used for immunoblotting included: rabbit polyclonal anti-Avidin antibody (1:5000, Abcam, Cambridge, UK; RRID: AB_305644); goat anti-rabbit peroxidase-conjugated secondary antibodies (1:10000, GE Healthcare). Blots were developed with the Luminata Forte Western Chemiluminescent HRP Substrate (Millipore, Burlington, MA, USA) on the ChemiDoc™ Imaging System (Bio-Rad). Densitometry was conducted with Image Lab 6.0 software (BioRad). The relative immune reactivity of the different proteins was normalized to the total protein loading by Ponceau Red staining (Fluka, Vancouver, BC, Canada).

### 2.8. Ex Vivo Fluorescence Imaging

Ex vivo optical imaging was performed on the excised brain and spinal cord after fluorescent NP administration. Fluorescence images were acquired with an IVIS Lumina III imaging system (PerkinElmer, Waltham, MA, USA). The following acquisition parameters were used: excitation filter range from 680 to 740 nm, emission filter (790 nm, exposure time 2 s), binning factor 4, and f/Stop 2. Spectral unmixing, image processing, and analysis were performed using Living Image 4.3.1 software (PerkinElmer).

### 2.9. Pharmacokinetics

#### 2.9.1. Sample Preparation and Extraction

The analytical method was revised in accordance with other protocols found in the literature [23,24]. As a first step, the internal standard (IS) fludrocortisone (10 ng) was added to the samples from treated mice, and Dex (0–300 ng) and fludrocortisone (10 ng) was included for the calibration curve. The liver, brain, and spinal cord tissue samples were treated with methanol (1:4 *w*/*v*) and acetonitrile (1:1 *w*/*v*), stirred, and sonicated for 20′. Then, water (1:10 *w*/*v*) was added, and the stirring and sonication steps were repeated. The resulting samples were centrifuged at 7000× *g* for 15 min at 4 °C. The supernatant was further purified with solid-phase extraction using Sep-Pak C18 1 cc Vac Cartridges (Waters, Milford, MA, USA) and was conditioned before use with 1 mL methanol, followed by 1 mL water. Samples were loaded on the SPE columns and passed through dropwise. Finally, the cartridges were rinsed with 1 mL water:acetone (80:20) and then with 1 mL of water before drying the columns under a vacuum for 5′. Samples were eluted with 1.8 mL acetonitrile into glass receiving tubes. Plasma aliquots were directly eluted with acetonitrile (1:4 *v*/*v*) and centrifuged at 7000× *g* for 15 min at 4 °C. All samples were evaporated to remove the organic phase. Just before analysis, they were suspended in 100 μL of 0.05% acetic acid:acetonitrile (80:20).

#### 2.9.2. Liquid Chromatography (HPLC) and Tandem Mass Spectrometry (MS/MS)

The LC–MS/MS system was a Nexera ultra-high-pressure liquid chromatography (UHPLC) system interfaced with a triple quadrupole LCMS-8060 (Shimadzu, Japan). The mass spectrometer (MS) operated in negative electrospray ionization (ESI) mode, with the following conditions: nebulizing gas flow rate 3 L/min, drying gas flow rate 10 L/min, heating gas flow rate 10 L/min, interface temperature 300 °C, and heating block temperature 400 °C. In a preliminary phase, standard solutions of Dex and the IS (100 ng/mL in water/acetonitrile) were directly injected into the mass spectrometer to identify and optimize the best ion transitions for MRM acquisition. The selected transitions and their collision energies (CE) were: m/z 361.2 > 292 (quantification transition, CE = 26) and m/z 361.2 > 325 (qualification transition, CE = 21) for Dex; m/z 349.1 > 295.2 (quantification transition, CE = 22) for IS. The chromatographic separation was obtained on an Ascentis C18 column (150 × 2.1 mm; 2.7-μm particle size, Sigma-Aldrich, St. Louis, MO) using an elution mixture composed of solvent A (0.05% acetic acid in water) and solvent B (acetonitrile) at 35 °C. The elution gradient was from 20 to 60% of solvent B in 12.5 min, from 60% to 99% of solvent B in 1.5 min (hold at 99% for 2 min), and re-equilibration in 4 min to 20% of solvent B. The injection volume was 5 μL, and the flow rate was 180 μL/min. Shimadzu’s LabSolutions software was used for instrument control, data handling, and analysis.

### 2.10. Statistical Analysis

Prism 7.0 (GraphPad Software Inc., San Diego, CA, USA) was used. For each variable, the differences between the experimental groups were analysed by one-way ANOVA followed by Tukey’s post hoc test. *p*-values below 0.05 were considered significant.

## 3. Results

### 3.1. ANANAS Formulations

The composition administered is shown in a schematic representation in Figure 1 and is summarized in Table 2.

The functional assemblies were generated starting from core ANANAS which had 25% of the available BBS covered with biotin-L-lys-(methyl-PEG5kDa) (biotin-(mPEG5kDa)_2_ [10]. This highly ω-methoxy-PEGylated core NP was selected based on recent results [15], which showed that a high number of ω-5 kDa PEG units at the outer NP surface gave them more stealth, favouring reaching out to tissues that were different than the liver, as for example the brain.

For ANANAS-ApoE-related investigations, following pre-formulation studies (Appendix A), we selected the NP composition with the maximum ApoE loading among the colloidally stable ones (10% BBS coverage). In the non-functionalized ANANAS, the surface of the core NPs was capped with biotin-methoxy-PEG5kDa (ANANAS-mPEG) to cover the same number of BBS (10%) as in ANANAS-ApoE. This formulation is also colloidally stable in the presence of biotin-C6-Alexa633, [13] which was added to cover 15% BBS for in vivo NP tracking with the IVIS instrumentation. On the contrary, alexa633 labelling of ANANAS-ApoE could not be performed due to an interference of the fully negatively charged dye with the positively charged ApoE component, which negatively affected the NP solubility.

The ANANAS-Dex assembly was generated by mixing core NPs with sub-saturating (57.5% of BBS) amounts of acid-sensitive short-spaced (C6) biotin-carbonate-hydrazone-dexamethasone conjugate (B-C6-Cb-Hz-Dex). The conjugate sub-saturating amount was selected so that no unbound conjugate remained in the assembly solution (see also Appendix A). Therefore, the latter could be administered as such without the need for further purification [10,11,13,15]. The biotin-C6-Cb-Hz-Dex conjugate used for dexamethasone loading was investigated in vitro in previous work [15] and was selected for the in vivo experiments of this work based on two main properties: (a) it is a low MW short-link spaced conjugate which permits high NP drug loading; (b) it is the least hydrolytically stable among the short-spaced Hz-dexamethasone conjugates investigated: a property which should favour fast drug release once it is internalized by target cells. The final formulation displayed in vitro a pH-dependent release of Dex, with no release at a neutral pH (as in serum) and 5.69% release/24 h at pH 4.0 [15].

### 3.2. In Vivo Studies

The experimental design developed for this study is reported in Figure 2.

To understand the impact of NP functionalization, a biodistribution study was performed in WT and SOD1^G93A^ mice, which were treated with ANANAS, ANANAS-ApoE or the vehicle and then sacrificed 30′, 4 h, and 24 h after administration (Figure 2A). As a proof of concept, SOD1^G93A^ mice were intravenously treated with Dex as a free molecule or linked to ANANAS (ANANAS-Dex) to evaluate the possible drug release from NPs. In this case, Dex was covalently linked to the ANANAS surface through an acid pH-sensitive linker [12,15] and should be released from the carrier upon reaching an acidic environment (cell internalization via endosome or an inflamed tissue). A pharmacokinetics study was performed by analysing tissue dex levels 15′, 30′, and 60′ after NP administration (Figure 2B). In addition to the brain and spinal cord, we analysed NP and drug levels at the liver as a control tissue since NPs introduced in the bloodstream are normally cleared by it and other organs of the reticuloendothelial system.

A methodological setting-up experiment was carried out in WT mice to evaluate the possible artefactual estimation of ANANAS accumulation inside the tissues due to blood contamination (Figure 3A). To this end, WT mice were injected intravenously with ANANAS. After 30 min of treatment, the first group was sacrificed and perfused with PBS to remove the blood, and the second one was sacrificed without perfusion. The liver, brain, and spinal cord were analysed for avidin levels. The liver showed the highest levels of NP-related avidin with no differences between the perfused and not perfused conditions. Probably, since the majority of NPs accumulate in the liver, the contribution of blood contamination is negligible. On the contrary, in the brain and spinal cord, where avidin levels are low, it is possible to appreciate a significant difference between perfused and non-perfused tissues. Therefore, we introduced the perfusion procedure in all further analyses.

We next moved to evaluate the presence and evolution over time of ANANAS and ANANAS-ApoE in the liver, brain, and spinal cord (Figure 3B–D).

In WT mice, ANANAS, both with and without ApoE, accumulated in the liver but with different kinetics. The non-functionalized ANANAS signal reached a significantly higher level versus the vehicle only 4 h after administration and further increased along the following hours. On the contrary, the level of ANANAS-ApoE was high already after 30 min, comparable to that of non-functionalized ANANAS at 24 h, and decreased along the following hours (Figure 3B). This indicates that the accumulation of ANANAS in the liver is favoured by the ApoE functionalization.

In the brain, the NP-related avidin signal was higher than the vehicle only in the ApoE-free ANANAS group. The levels were the highest at 30 min, then significantly decreased over 24 h. On the other hand, levels of NP in ANANAS-ApoE-treated mice were similar to those of the vehicle group, indicating that the functionalized NP was unable to penetrate the brain (Figure 3C).

In the spinal cord mice treated with both ANANAS and ANANAS, ApoE showed NP-related avidin levels above the vehicle but with a different trend. In particular, only non-functionalized ANANAS reached significant levels in the tissue 30 min after administration and then decreased in the next hours. ANANAS-ApoE levels were slightly higher than the controls; however, this was without reaching significance (Figure 3D), and this did not change over time. These results show that ANANAS can reach the spinal cord, but the presence of ApoE has a negative impact on this property.

Overall, biodistribution data in WT mice indicate that, in the early phase, the unmodified ANANAS can reach both the spinal cord and brain with similar kinetics and accumulates in the liver over the next 24 h. The presence of ApoE prevents ANANAS accumulation in CNS tissues and favours NP accumulation in the liver already from the early time points.

Tissue biodistribution analysis of NPs over time was also performed in the ALS disease animal model (Figure 4).

SOD1^G93A^ mice were treated with ANANAS and ANANAS-ApoE at the onset of symptoms (14 weeks of age): a stage at which treatments are usually started in preclinical drug testing and when BBB and BSCB damage is normally observed [14,25].

In the liver, NP levels were already high 4 h after administration and independently of the presence of ApoE at the NP surface, confirming the strong tropism of the ANANAS carrier for this tissue. Similar to what was observed in WT time, the presence of ApoE was correlated with a faster NP signal decay from this tissue. The levels of untargeted ANANAS remained stable between 4 and 24 h and decreased by about 30% in the case of the ApoE-linked formulations (Figure 4A).

Thirty min after administration, NP-related avidin levels in the brain were just above the controls in ANANAS-ApoE-treated mice. On the other hand, levels were significantly higher in animals treated with untargeted ANANAS, whose signal slowly decreased over time, reaching that of the ANANAS-ApoE group (Figure 4B) 4 h after administration.

In the spinal cord, NP-related avidin was already detectable 30 min after administration in ANANAS-treated animals, and again, the levels gradually decreased over time, reaching untreated control levels 4 h after administration. In the case of the ANANAS-ApoE-treated mice, the avidin-related signal was barely detectable (without significance) at all time points. The results also confirmed in the SOD1^G93A^ mouse (Figure 4C) the ability of non-functionalized NPs to reach the spinal cord earlier and with greater amounts than ApoE-NPs.

Since functionalization with ApoE did not induce an improvement in ANANAS penetration inside the brain and spinal cord in either healthy or ALS mice, we focused our attention on non-functionalized ANANAS in SOD1^G93A^ mice. A biodistribution study 30′ and 24 h after ANANAS intravenous administration was carried out using ex vivo optical imaging (IVIS Lumina XRMS) (Figure 5).

For longitudinal tracking, ANANAS were labelled with the fluorophore biotin-alexa633, which was added to cover 5% of the total biotin binding sites available. The ex vivo analysis showed that at both 30 and 24 h after administration, the fluorescent signal associated with ANANAS was detectable in both the brain and spinal cord, with significant differences with respect to the vehicle group at 30′ and 24 h in the spinal cord and the brain, respectively. Figure 5A shows a rapid increase in the signal associated with the administration of ANANAS compared to the control mouse. In the SOD1^G93A^ mouse model, motor neuron degeneration affects mostly the lumbar spinal cord. For this reason, we measured the lumbar and cervical regions separately in case degeneration affected CNS penetration. However, our data indicate no anatomical differences and, therefore, a homogeneous distribution of the nanoformulation throughout the spinal cord. Thus, in the histogram, we reported the fluorescent signal to be associated with the entire cord.

In recent decades, several drugs have been tested in preclinical ALS models targeting different molecular mechanisms with very limited results. One of the possible explanations for this failure is that the majority of the available compounds have poor BBB permeability. On the other hand, a proof-of-concept study provided encouraging results through the invasive intracranial delivery of a promising anti-inflammatory compound [25]. It is, therefore, pivotal to investigate innovative methods for the release of compounds with therapeutic potential. Therefore, we evaluated the combination of the CNS-permeable ANANAS with an established anti-inflammatory drug, such as dexamethasone.

Figure 6 shows the results of the pharmacokinetics study carried out in SOD1^G93A^ mice treated intravenously with ANANAS-Dex at the onset of symptoms.

Similar to what was reported in healthy mice treated with ANANAS-Dex formulations, in the SOD1^G93A^ model, the majority of the free Dex was detected in the liver, whereas only low levels were found in the plasma (from 2 to 1 ng/mL over time, Figure 6A). This is explainable by the fact that, in these formulations, the drug is linked through an acid reversible linker to be released after the localization of the ANANAS in an acidic compartment (e.g., the liver macrophage endo/lysosomes or an inflamed tissue). Unfortunately, no free Dex was detected in the brain and spinal cord homogenates (Figure 6B) of the SOD1^G93A^ mice. This probably indicates that in these tissues, the conditions favouring the release of the drug from the linker were not met.

## 4. Discussion

The BBB is considered the major obstacle to the pharmacological treatment of CNS disorders. The tight junctions, together with the activity of several transporters at the barrier tissues, hinder penetration in CNS and the persistence of a very large amount of substances and drugs. More than 98% of small molecule candidate drugs and the majority of novel proteins and peptides are unable to reach the brain tissue due to poor permeability across the BBB [26]. Nanotechnologies through engineered and functionalized NPs may represent a suitable strategy to overcome this challenge. In this study, we detected the presence of ANANAS in the brain and spinal cord of healthy and SOD1^G93A^ mice, especially in the first interval of time after administration (30′ and 4 h), since an active process of clearance occurs 24 h after NP injection. Surprisingly, we found no signal related to the NPs in the brain and spinal cord of mice treated with ANANAS-ApoE. Such a paradoxical behaviour is not in agreement with what has been previously reported, especially in vitro, and supported the exploitation of the lipoprotein-associated receptor transport as an attractive strategy to shuttle drug molecules across the BBB. In particular, the low-density lipoprotein receptors (LDLR), characterized by a high affinity for ApoE, seemed functional to this aim. On the other hand, more recent in vivo data clearly demonstrated that, once in the bloodstream, NPs underwent a series of events that dramatically hamper the brain endothelial targeting and the possibility to cross the BBB. In addition to the uptake by organ filters and renal clearance, circulating NPs are functionalized to enhance specific targeting and tend to lose their affinity over time. This reduced ability to bind the appropriate ligands or receptors is due to the coverage of the NP surface by circulating plasma proteins through a process widely known as “protein corona” [27]. The effect of plasma protein interaction with NPs is still a matter of debate. If there are much data confirming the shielding effect played by protein corona [28], on the other hand, a pre-formed, homogeneous protein corona around NPs using only ApoE increased the uptake by cancer cells overexpressing LDLR [29]. However, the efficacy of the BBB passage of this naturally functionalized complex is not known. A possible explanation for the poor BBB passage of our ANANAS-ApoE can be related to endogenous ApoE expression and its interaction with LDLR. LDLR is expressed in the liver, peripheral vasculature, brain, and other tissues [30,31], and in the brain is the most abundantly expressed neuron. Endogenous ApoE is primarily produced by glial cells [32]; therefore, we can hypothesize that astroglial ApoE can compete and interfere with the ligand exposed to the ANANAS surface and limit its CNS penetration.

Disease condition and progression have to be considered in NP design and CNS penetration. BBB/BSCB functional and structural impairments in neurological disorders have been reported in several studies [2]. In both patient and animal ALS models, damage in BBB/BSCB integrity and function, the downregulation of tight junction proteins, endothelial cell degeneration, and impairments in the micro vessels [3,33] have been observed. This suggests that in the pathological condition, BBB/BSCB may be more permeable to drugs. However, our results show the same kinetics of ANANAS for brain and spinal cord tissue in healthy and SOD1^G93A^ mice. Non-functionalized ANANAS in both animal models shows the major penetrance in the brain and spinal cord immediately after NP injection, suggesting an equal CNS penetrance of NP in healthy and diseased mice. It has been reported that in ALS patients and in SOD1^G93A^ mice, the P-glycoprotein and breast cancer resistance protein, two drug efflux transporters at the brain endothelium with remarkably broad specificity, increase their expression and function at the BBB and BSCB as the disease progresses [34]. Therefore, it is possible to hypothesize that despite the potential breakage of BBB, a disease-driven process of pharmaco-resistance limits the ability of the NPs to pass the BBB.

When using NPs for brain drug delivery, the first question that has to be answered is whether, after their passage through the BBB, they can reach the pathological target and selectively release the cargo. However, increased NP concentration in the brain (as in the case of ANANAS) does not necessarily imply that a selected drug load can be released in the affected areas. To better investigate the relationship between the carrier and cargo, we tested a Dex carrying an ANANAS formulation, in which the steroid was linked to the NPs by means of a pH-sensitive reversible linker. In previous work by our group, an ANANAS-Dex formulation with a strong tropism for the liver showed therapeutic efficacy in a murine model of autoimmune hepatitis [12,15]. A remarkable characteristic of ALS patients and animal models is the neuroinflammatory reaction consisting of activated glial cells and T cells which have been largely explored as a potential therapeutic target. In particular, free Dex and other corticosteroids have shown encouraging neuroprotective and anti-inflammatory effects in the mouse models of motor neuron diseases [16,17]. However, none have translated to an effective therapy in patients, probably because of their side effects [35,36,37]. Targeting steroids into the CNS by NPs may be a way to maximize the therapeutic effect while reducing unwanted side effects [38]. In the case of ANANAS, the strong tropism for CD68-positive phagocytic cells, in particular for those carrying Dex [12], was considered attractive for ALS and aimed to reduce microglial cell activation. Unfortunately, Dex levels that were released from NPs in the brain and spinal cord were under the detection limit. There are several possible reasons why the ANANAS penetration did not correspond to a detectable amount of Dex. The more probable is that the conditions that allowed the release of the steroid from the acid pH-sensitive linker were not met as it otherwise occurred in the lysosomes of the liver macrophages reached by the ANANAS-Dex formulation investigated in our previous studies [12,15].

In conclusion, in the present study, we demonstrated that NP penetrance in CNS is not sufficient to enable efficient drug brain- and spinal cord-targeting. Specifically, the hydrazone bond used here for dexamethasone tethering to ANANAS was clearly not suitable for the release of the drug to the brain tissue. Therefore, in order to fully take advantage of the NP’s ability to penetrate the CNS, a different strategy for drug tethering should be devised, which does not require the NP cell internalization for drug release but rather exploits brain parenchyma-specific enzymatic activities. On the other hand, this study demonstrates that drug measurement studies in vivo are a necessary step in the development of a nano-drug. Moreover, it highlights that NPs can be useful for improving the passage through biological barriers but may limit the interaction of the therapeutic compound with the specific target.

## Figures and Tables

**Figure 1 cells-11-04003-f001:**
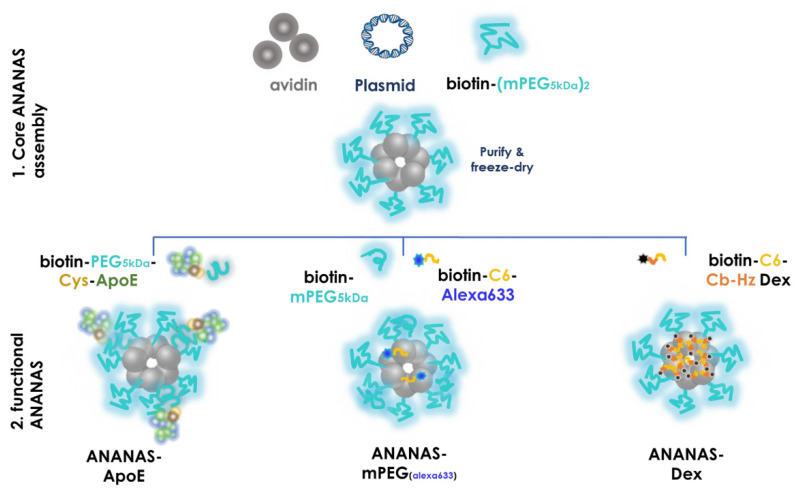
Schematic representation (not in scale of the formulations selected for the in vivo studies). Upper panel: components of the core ANANAS; lower panel: functional assemblies are generated by mixing core ANANAS with the different biotin components at predefined biotin:ANANAS BBS molar ratios.

**Figure 2 cells-11-04003-f002:**
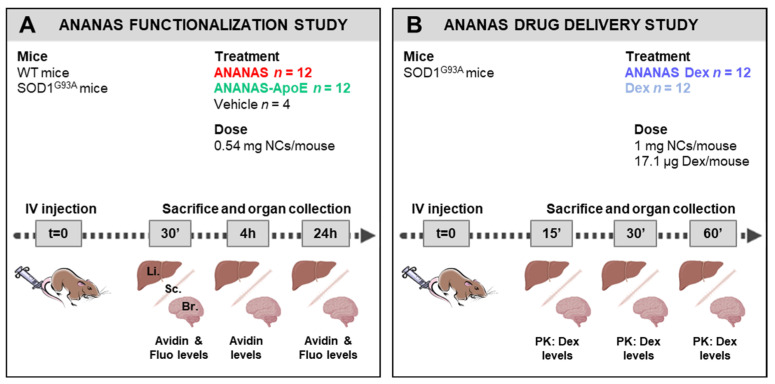
Experimental plan for NP functionalization and delivery study (**A**) WT and SOD1^G93A^ mice (*n* = 12 mice per group) were intravenously (IV) treated with ANANAS, ANANAS-ApoE, or vehicle for ANANAS functionalization study. Liver (Li.), spinal cord (Sc.), and brain (Br.) were collected 30 min, 4 h and 24 h post treatment for biochemical and imaging analysis. (**B**) SOD1^G93A^ mice (*n* = 12 mice per group) were intravenously (IV) treated with ANANAS-Dex or Dex for ANANAS drug delivery study. Organs were collected for PK measurements 15-, 30-, and 60-min post treatment.

**Figure 3 cells-11-04003-f003:**
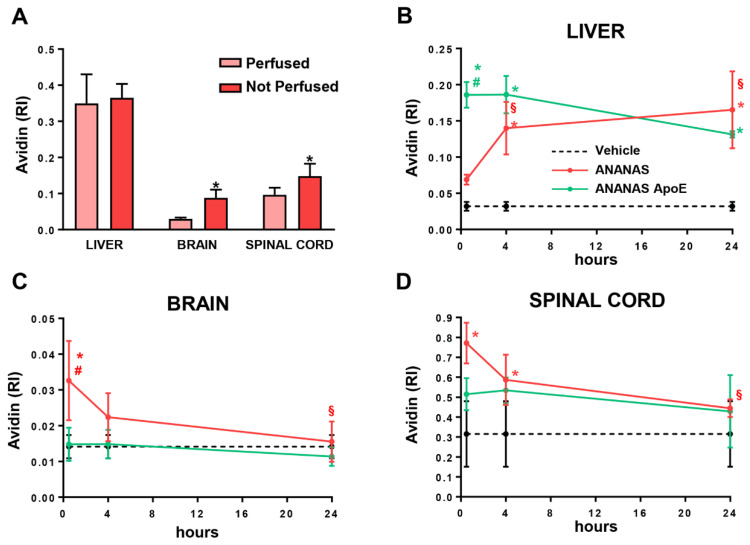
NP biodistribution in WT mice. (**A**) Avidin quantification from liver, brain, and spinal cord of WT mice intravenously treated with ANANAS and sacrificed 30 min after injection, with (pink bars) or (red bars) without intracardial perfusion. Data are mean ± SEM (*n* = three in each experimental group) and indicates the relative immunoreactivity (RI) normalized to total protein loading; * *p* < 0.05 versus the respective perfused tissue by Student’s *t*-test. (**B**–**D**) Dot blot analysis for avidin quantification in liver (**B**), brain (**C**), and spinal cord (**D**) of WT mice treated with vehicle (dotted line), ANANAS (red line), and ANANAS-ApoE (green line) and sacrificed with intracardial perfusion 30 min, 4 h, and 24 h after the injection. (**B**–**D**) Data are mean ± SD (*n* = three or four in each experimental group) and indicates the relative immunoreactivity (RI) normalized to total protein loading; * *p* < 0.05 versus vehicle; # *p* < 0.05 versus ANANAS-ApoE and § *p* < 0.05 versus ANANAS 30 min, by one-way ANOVA, Tukey’s post hoc test.

**Figure 4 cells-11-04003-f004:**
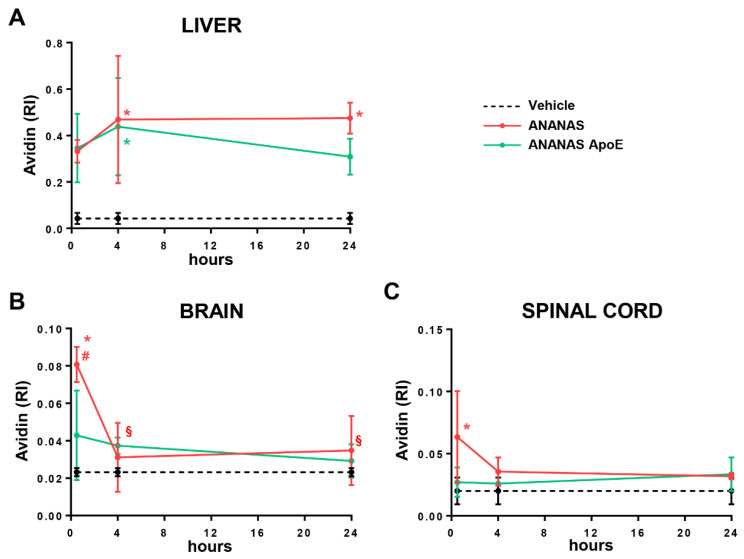
NP biodistribution in SOD1^G93A^ mice. (**A**–**C**) Dot blot analysis for avidin quantification in the liver (**A**), brain (**B**), and spinal cord (**C**) of SOD1^G93A^ mice treated with a vehicle (dotted line), ANANAS (red line), and ANANAS-ApoE (green line) and sacrificed with intracardial perfusion 30 min, 4 h, and 24 h after the injection. (**A**–**C**) Data are mean ± SD (*n* = three or four in each experimental group) and indicates the relative immunoreactivity (RI) normalized to total protein loading; * *p* < 0.05 versus vehicle; # *p* < 0.05 versus ANANAS-ApoE and § *p* < 0.05 versus 30 min, by one-way ANOVA, Tukey’s post hoc test.

**Figure 5 cells-11-04003-f005:**
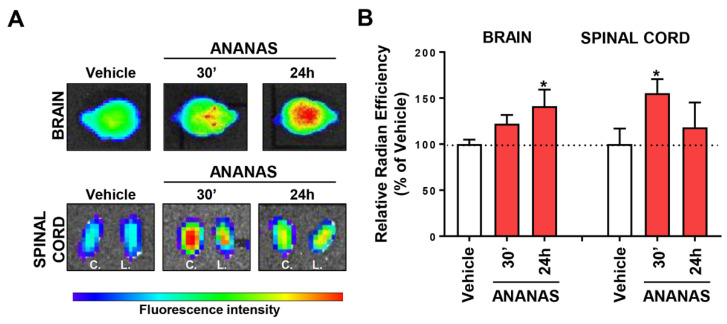
Ex vivo biodistribution in SOD1^G93A^ mice. (**A**,**B**) Ex vivo optical analysis of brain and spinal cord from SOD1^G93A^ mice treated with saline (Vehicle) or with ANANAS and sacrificed with intracardial perfusion 30 min and 24 h after the treatment. (**A**) Representative images of brain and cervical (C.), lumbar (L.), spinal cord. (**B**) Quantification of ex vivo optical imaging signal in brain and spinal cord. Dotted lines indicate the mean of vehicle. Data (mean ± SD; *n* = three or four in each experimental group) indicates the relative radian efficiency and are expressed as percentages of the respective vehicle. * *p* < 0.05 versus vehicle, by one-way ANOVA, Tukey’s post hoc test.

**Figure 6 cells-11-04003-f006:**
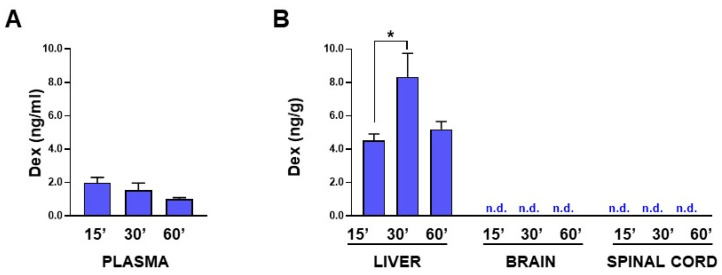
Pharmacokinetics study of free Dex in SOD1^G93A^ mice. Levels of free Dex measured in plasma (**A**) and liver, brain, and spinal cord (**B**) from SOD1^G93A^ mice at different time points (15, 30, 60 min) after intravenous administration of ANANAS-Dex. Dex released from the NP was measured by HPLC MS/MS with limit of quantitation (LoQ) equal to 0.01 ng/g for the brain and spinal cord, and 0.1 ng/mL for plasma. Data are mean ± SEM (*n* = four in each experimental group); * *p* < 0.05 by one-way ANOVA, Tukey’s post hoc test. n.d.: not detected.

**Table 1 cells-11-04003-t001:** Animals, treatment schemes, and type of analysis.

Mice	Intravenous Treatment	Time of Sacrifice	Perfusion	Analyses
4 WT	ANANAS	30′	Yes	Dot blot
4 WT	ANANAS	30′	No	Dot blot
4 WT	Vehicle	24 h	Yes	Dot blot
4 WT	ANANAS	30′	Yes	Dot blot
4 WT	ANANAS	4 h	Yes	Dot blot
4 WT	ANANAS	24 h	Yes	Dot blot
4 WT	ANANAS-ApoE	30′	Yes	Dot blot
4 WT	ANANAS-ApoE	4 h	Yes	Dot blot
4 WT	ANANAS-ApoE	24 h	Yes	Dot blot
4 SOD1^G93A^	Vehicle	24 h	Yes	ex vivo imaging, Dot blot
4 SOD1^G93A^	ANANAS	30′	Yes	ex vivo imaging, Dot blot
4 SOD1^G93A^	ANANAS	4 h	Yes	ex vivo imaging, Dot blot
4 SOD1^G93A^	ANANAS	24 h	Yes	ex vivo imaging, Dot blot
4 SOD1^G93A^	ANANAS-ApoE	30′	Yes	Dot blot
4 SOD1^G93A^	ANANAS-ApoE	4 h	Yes	Dot blot
4 SOD1^G93A^	ANANAS-ApoE	24 h	Yes	Dot blot
4 SOD1^G93A^	ANANAS-Dex	15′	Yes	HPLC/MS, Dot blot
4 SOD1^G93A^	ANANAS-Dex	30′	Yes	HPLC/MS, Dot blot
4 SOD1^G93A^	ANANAS-Dex	60′	Yes	HPLC/MS, Dot blot
4 SOD1^G93A^	Dex	15′	Yes	HPLC/MS, Dot blot
4 SOD1^G93A^	Dex	30′	Yes	HPLC/MS, Dot blot
4 SOD1^G93A^	Dex	60′	Yes	HPLC/MS, Dot blot

**Table 2 cells-11-04003-t002:** Composition and size of the formulations used in the in vivo studies.

Formulation Name	Biotin Reagent Added to Core NPs	(% BBS)	Z-Average (nm) 1 h	PDI (1 h)	Z-Average (nm) 24 h	PDI 24 h	ζ Potenzial
ANANAS-ApoE	B-PEG5kDaCys-ApoE	10	138.9 ± 0.3	0.26	142.4 ± 7.0	0.36	−3.76 ± 0.903
ANANAS-mPEG	mPEG5kDa (*)	10	136.9 ± 1.5	0.17	137.3 ± 2.5	0.23	−6.50 ± 0.310
ANANAS-Hz-Dex	BC6-Cb-Hz-Dex	57.5	141.1 ± 1.1	0.31	142.5 ± 2.1	0.33	−5.8 ± 1.711

(*) for in vivo fluorescent tracking, biotin-C6-Alexa633 was added to cover 15% BBS.

## Data Availability

The datasets presented in this study can be found in online repositories. The names of the repository/repositories and accession number(s) can be found below: 10.5281/zenodo.6504947.

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
