# Peer review of "Development of a Nanoparticle-Based Approach for the Blood–Brain Barrier Passage in a Murine Model of Amyotrophic Lateral Sclerosis"

_cells, 2022, doi:10.3390/cells11244003_

Round 1
Reviewer 1 Report
the article is well written. all the figures are will presented. cross check the references in txt and references.
Author Response
We thank the reviewer for the positive comments on the manuscript. As suggested, the main text was revised by a native English speaker and majorly edited for clarity.
Reviewer 2 Report
1. The article is written badly and there are many typos and errors. There are flaws in English. I would highly recommend to publish only when English corrections and typos as well as errors are rectified. The greatest example in the discussion section is, " Brain of Spinal Cord" and many more illogical phrases are present
2. ANANAS nanoformulation with Dexamethasone can be explained in details as to why this study has been done and whats the sigificance of Dexamethasone in this study. Please also add relevant references too.
3. Its been shown here that the release of drug takes place majorly in the plasma, what is the release mechanism of the drug?
4. Can ANANAS be explained in details, such as what kind of plasmid DNA is used?
Author Response
We take this opportunity to thank the reviewer for the careful revision of our paper, as the suggestions have significantly contributed to improve the quality of the manuscript. Please, find below in the attachment a point-by-point response to the issues raised by the reviewer.
